# Effects of an Unripe Avocado Extract on Glycaemic Control in Individuals with Obesity: A Double-Blinded, Parallel, Randomised Clinical Trial

**DOI:** 10.3390/nu15224812

**Published:** 2023-11-17

**Authors:** Lijun Zhao, Donald K. Ingram, Eric Gumpricht, Trent De Paoli, Xiao Tong Teong, Bo Liu, Trevor A. Mori, Leonie K. Heilbronn, George S. Roth

**Affiliations:** 1Adelaide Medical School, The University of Adelaide, Adelaide, SA 5005, Australia; lijun.zhao@adelaide.edu.au (L.Z.); xiaotong.teong@adelaide.edu.au (X.T.T.); bo.liu14@yahoo.com (B.L.); leonie.heilbronn@adelaide.edu.au (L.K.H.); 2Lifelong Health Theme, South Australian Health and Medical Research Institute, Adelaide, SA 5000, Australia; 3Pennington Biomedical Research Center, Louisiana State University, Baton Rouge, LA 70808, USA; 4Isagenix International LLC, Gilbert, AZ 85297, USA; eric.gumpricht@isagenixcorp.com; 5De Paoli Farms, Bundaberg, QLD 4670, Australia; trent@farmerceuticals.com.au; 6Discipline of Internal Medicine, Medical School, University of Western Australia, Perth, WA 6000, Australia; trevor.mori@uwa.edu.au; 7Prolongevity Technologies LLC, Pylesville, MD 21132, USA; geor@iximd.com

**Keywords:** avocado, insulin, mannoheptulose, glucose, cardiovascular risk factors, autophagy

## Abstract

Background: Unripe avocados (*Persea americana*) are naturally enriched in mannoheptulose (MH), which is a candidate caloric restriction mimetic. Objectives: To evaluate the effects of a diet supplement made from unripe avocado on glucose tolerance, and cardiometabolic risk factors in free-living nondiabetic adults with obesity. Methods: In a double-blinded, randomised controlled trial, 60 adults (female n = 47, age 48 ± 13 years, BMI 34.0 ± 2.6 kg/m^2^) were stratified by sex and randomised to avocado extract (AvX, 10 g finely ground, freeze-dried unripe avocado) or placebo (10 g finely ground cornmeal plus 5% spinach powder) daily, for 12 weeks. The primary outcome was a change in glucose area under the curve (AUC) in response to a 75 g oral glucose tolerance test. A post-hoc analysis was subsequently performed in a subgroup with insulin AUC above the median of baseline values after removal of participants >2 SD from the mean. Results: There were no between-group differences in glucose AUC (*p* = 0.678), insulin AUC (*p* = 0.091), or cardiovascular outcomes. In the subgroup analysis, insulin AUC was lower in AxV versus placebo (*p* = 0.024). Conclusions: Daily consumption of unripe avocado extract enriched in MH did not alter glucose tolerance or insulin sensitivity in nondiabetic adults with obesity, but the data provided preliminary evidence for a benefit in insulin AUC in a subgroup of participants with elevated baseline postprandial insulin levels.

## 1. Introduction

Obesity is a global concern associated with adverse health consequences, which include increased risk of cardiovascular diseases (CVD), diabetes mellitus and cancer [1,2,3]. Moderate calorie restriction (CR) is effective in reducing body weight and the risk of chronic disease [4]. As a result of inherent body defence networks that reduce metabolic rate [5] and increase hunger and appetite [6], CR is difficult to maintain long-term [7]. There are also some side effects including reductions in libido and bone mineral density [8]. Thus, there is significant interest in the identification of novel CR mimetics, which mimic the metabolic and physiological benefits of CR, without requiring a reduction in food intake [8]. Several pathways involved in response to CR have been proposed as targets for mimetics, and candidate molecules are being investigated [8]. The current study emerged from interest in developing inhibitors of intracellular glycolysis to stimulate CR-like responses [8].

Mannoheptulose (MH) is a seven-carbon monosaccharide that naturally occurs in unripe avocados (*Persea americana*) as well as other plants, such as primrose and alfalfa [9]. While avocados have relatively high levels of MH compared to other fruits, the highest concentrations of this rare sugar are found in unripe avocados with only trace levels found in some ripe varieties of this fruit [9]. MH is known to be a glycolytic inhibitor acting on hexokinase, the first step in intracellular glycolysis [8]. MH has been proposed as a CR mimetic since it increased the median and maximal lifespan in *Drosophila melanogaster* by 15% [9,10]. L6 myocyte cultures treated with MH showed a dose-dependent increase in 5′ adenosine monophosphate-activated protein kinase, NAD-dependent deacetylase sirtuin-1 and peroxisome proliferator-activated receptor γ coactivator 1-α protein levels consistent with the molecular signalling profile of CR [11]. Dietary supplementation with MH via an extract from unripe avocados diminished the adverse effects of a high-fat diet in mice, including increased insulin sensitivity, improved lipid profile, reduced reactive oxygen species and extended median lifespan in mice fed a chow diet [10,12]. Consistent with a profile of CR, supplementation with an MH-enriched extract of unripe avocados reduced oxidative stress and inhibited the mTOR pathway to activate cellular autophagy in mice fed a high-fat diet [12]. Dogs receiving 2–20 mg/kg of MH via extract of unripe avocados exhibited reduced fasting insulin in a dose-dependent manner [11,13]. However, a second study in dogs could not replicate these findings [14], but increases in fasting levels of glucagon-like peptide-1 and post-prandial levels of ghrelin were observed, which are suggestive of an appetite suppressant effect [15].

Human studies describing consumption of MH are extremely limited. To our knowledge, only one previous study in humans administered MH contained in avocados (at a dose of 2–13 g), to eight men [16]. In that study, fasting insulin was reduced in five of the eight men fed the avocados, with no discernible change in plasma glucose [16].

The present study evaluated the effects of supplementing the diet for 12 weeks with 10 g of extract made from dried, unripe avocados that are naturally high in MH, versus a placebo made from cornmeal, on glycaemia and cardiometabolic risk factors in nondiabetic humans with obesity. The primary outcome was glucose area under the curve (AUC) in response to an oral glucose tolerance test (OGTT). Secondary outcomes were changes in insulin AUC, insulin resistance assessed using the Homeostatic Model Assessment for Insulin Resistance (HOMA-IR), β-cell function (HOMA-β), Matsuda Insulin Sensitivity Index, cardiometabolic risk markers and autophagy gene expression.

## 2. Materials and Methods

### 2.1. Participants

The study employed a two-arm, parallel group, double-blinded, randomised controlled trial with 60 participants recruited between 9 March 2021 and 11 August 2022. The study was conducted according to the guidelines of the Declaration of Helsinki, and approved by the Human Research Ethics Committee of The University of Adelaide (protocol code H-2020-248 on 17 November 2020) and registered on ClinicalTrials.gov (NCT04763473). Written, informed consent was obtained from all participants prior to their enrolment. Inclusion criteria were as follows: men and women aged 40–70 years; body mass index (BMI) between 30 and 40 kg/m^2^; waist circumference ≥94 cm for men and >80 for women; weight-stable (<5% fluctuation in body weight for 6 months prior to study entry); willingness to provide written informed consent; and willingness to participate and comply with the study (Figure 1).

Participants were excluded if they reported any of the following: weight change >5% fluctuation in their body weight for 6 months before study entry; personal history of bariatric surgery; diagnosis of type 1 or type 2 diabetes mellitus, liver or kidney diseases, neoplastic disease in the previous three years, chronic gastrointestinal disorders (including inflammatory bowel disease and celiac), cardiovascular event in the previous 6 months, biochemical abnormalities or evidence at screening of disease including elevated liver enzymes ALT and/or AST >3 times normal range limit; current or recent (within 12 months) treatment with medication used to lower blood glucose or antidiabetic medications (metformin, sulfonylureas, glucagon-like peptide-1 (GLP-1) analogues [i.e., exenatide], thiazolidinediones or DPP-IV inhibitors [i.e., ‘gliptins’]), medications affecting weight, appetite or gut motility (i.e., domperidone, cisapride, orlistat, phentermine, topiramate); women planning pregnancy during the course of the study or three months after completion of the study, or who were lactating; and conditions that could interfere with the ability to understand the requirements of the study. Participants who were taking stable doses (i.e., >12 months) of androgenic medications (i.e., testosterone), thyroxine, anti-depressants (selective serotonin reuptake inhibitors), anti-hypertensives (ace-inhibitors, calcium channel blockers, beta-blockers, diuretics) and lipid-lowering medications (statins, fibrates) were not excluded.

### 2.2. Study Design

The trial period was 13 weeks, including 1-week baseline monitoring and a 12-week intervention. First-pass eligibility testing was performed by email/phone with a screening questionnaire according to the inclusion/exclusion criteria detailed above. Participants who appeared eligible were invited to attend a screening visit at South Australian Health and Medical Research Institute. Participants had the research protocol explained and provided the written consent form to participate in the study. Weight, height, and blood pressure were measured. A fasting blood sample was collected to evaluate liver (liver enzymes >3 times upper limit considered normal) and kidney (creatinine and estimated glomerular filtration rate <60 mL/min/1.73 m^2^) function and whole blood count. If eligible, the participants were enrolled in the study. For one week prior to baseline (Week 1) and week 12 (Week 11), participants were fitted with an ActiGraph to measure daily activity levels, step counts and sleep patterns (wGT3X-BT, ActiGraph LLC, Pensacola, FL, USA) and asked to record all eating and drinking events via smartphone application (Research Food Diary, Xyris Software Pty Ltd., Sydeny, Australia). Metabolic tests were performed at week 0 and week 12 of the intervention. During the metabolic visit, participants arrived at our research facility at 08:00 h after fasting from 20:00 h. Body weight was measured in a hospital gown after voiding, and blood pressure was measured after a 10-min seated rest. A 20-gauge cannula was inserted into an antecubital vein of the nondominant arm, and a fasting blood sample was collected. Following a 75 g oral glucose drink, blood samples were collected at 0, 30, 60, 90, and 120 min. Participants also visited the South Australian Health and Medical Research Institute (SAHMRI) clinic room every four weeks for a check-in visit to assess adherence and any changes in health, to return supplement bags for weighing as a measure of compliance and to receive more supplements as needed.

### 2.3. Supplement Preparations

Whole avocado fruit of the Hass variety was obtained from an Australian commercial crop (De Paoli Farms, Queensland, Australia) in an unripe state and stored in a cold room under 5 °C. Within 24 h after the harvest, the whole fruit was washed with water, and the flesh and peel were commercially shredded by a disintegrator, before being transferred into a freeze-drying machine for 96 h. The mass percentage of MH in avocado extract (AvX) was 1.9%, which was determined by HPLC at Southern Cross University (T Block Level 3, Military Road, Lismore, NSW, Australia) according to the methods developed by Kappler-Tanudyaya et al. [17] and Shaw et al. [18]. The placebo was a commercially available precooked fine-ground corn meal (~37 kCal, PANCORN, USA) that was spiked with 5% spinach powder (Australian Spinach Vegetable Powder, Boost Nutrients, NSW, Australia). Both products (180 g) were transferred in identical heat-sealable foil bags. The bags were coded red or blue by an individual who was not involved in the study. Compliance was assessed by weight of the retuned product minus initial weight. Both groups were highly compliant with 85 ± 16% of the prescribed amount consumed in the Avocado group and 79 ± 9% of the prescribed amount consumed in the placebo groups, respectively. Based upon calculations from previous dog studies revealing beneficial effects of an MH enriched extract of unripe avocado on glucose/insulin responses, our target dose of MH was 2 mg/kg [11]. Since 2 mg/kg was at the lower end of dose effectiveness, we had originally planned for another treatment group provided a higher dose, e.g., 10–20 mg/kg; however, the calculated yield of the extract did not allow inclusion of a high-dose group.

### 2.4. Dietary Intervention and Randomisation

Following metabolic testing at week 0, 60 participants were randomised with a 1:1 ratio to red (placebo) or blue (avocado) for 12 weeks. Stratified blocked randomisation by sex was employed using a web-based system under the direction of LKH, who had no contact with participants. This study was double-blinded; neither participants nor study personnel knew the randomisation scheme. Both groups were instructed to consume one scoop (~10 g) of AvX or placebo, daily, with breakfast. No other dietary instructions were provided, and participants were asked to maintain their usual daily physical activity and sleep patterns throughout the study.

### 2.5. Outcome Measures

The primary outcome measure was the change in glucose area under the curve (AUC) in response to a 75 g OGTT at week 12. The secondary outcomes included changes in body weight, body composition, systolic (SBP) and diastolic blood pressure (DBP), fasting blood lipids, high sensitive C-reactive protein (hsCRP), fasting glucose and insulin, insulin AUC in response to the OGTT, HOMA-IR, HOMA-β, Matsuda Insulin Sensitivity Index, oxidative stress (plasma F2-isoprostanes) and blood mRNA markers of autophagy (*LC3*, *BECN1*, *P62*, *LAMP2* and *TFEB*).

### 2.6. Body Composition

During each metabolic visit, body weight and waist circumference were measured with participants in a gown after voiding. Body weight was measured to the nearest 0.1 kg with a calibrated scale (Tanita BWB-800 digital scale, Wedderburn Pty Ltd., Sydney, Australia). Body mass index was calculated as weight in kilograms per height in meters squared. Body fat was assessed by Dual-energy X-ray absorptiometry (DEXA, Lunar Prodigy; G.E. Health care, Madison, WI, USA) and analysed by enCore software (G.E. Healthcare, Chicago, IL, USA; version 18).

### 2.7. Blood Pressure

SBP and DBP were measured twice with the participant seated after 10 min of rest (Welch Allyn, NY, USA). A third measurement was taken if there was a difference of >3 mm/Hg. The mean of the two lowest blood pressure readings was used.

### 2.8. Dietary Intake

Participants were asked to self-report all their dietary intake via a smartphone application (Easy Diet Diary, Xyris Software (Australia) Pty Ltd., Sydney, Australia) one week prior to each metabolic testing at baseline and week 12. Adherence to the diets was assessed by a research dietitian using the participants’ self-reported seven-day food records. The energy and macronutrient intakes were calculated by using FoodWorks Professional (version 10; Xyris Software (Australia) Pty Ltd., Australia). The leftover of the supplements at each visit was weighed to calculate the total amount consumed.

### 2.9. Physical Activity

Participants were fitted with a waist-worn triaxial accelerometer (ActiGraph wGT3X-BT, LLC, Pensacola, FL, USA) using an elastic belt to secure above the right hip bone to measure their movements and activities for at least 7 days prior to each metabolic testing. The monitor was initialized at a sample rate of 30 Hz to record activities for free-living conditions. ActiGraph data were downloaded and analysed by using ActiLife 6 software after collecting the devices from participants.

### 2.10. OGTTs

Metabolic tests were conducted at week 0 and 12 in all groups. Participants arrived at 07:30 h at the Research Unit following a 12 h overnight fast. A cannula was inserted for collection of fasting blood (30 mL) immediately prior to consuming a 75 g oral glucose drink (Carbotest, POCD scientific). Participants consumed the glucose drink within 5 min. Blood samples were collected at 15, 30, 60, 90, 120 and 150 min after consumption. Serum and plasma were immediately placed on ice, centrifuged at 4 °C, and frozen to −80 °C for later analysis.

### 2.11. Blood Analysis

Plasma glucose, total cholesterol, high-density lipoprotein-cholesterol (HDL-C) and triglycerides and hsCRP were measured using commercially available enzymatic kits on an automated clinical analyser (INDIKO Plus, Thermo Fisher Scientific Inc., Vantaa, Finland). Low-density lipoprotein-cholesterol (LDL-C) was calculated according to the Friedewald equation: LDL-C = (Total Cholesterol − HDL-C) − (triglycerides/5) [19]. Plasma insulin was measured by radioimmunoassay (HI-14K, Millipore, Burlington, MA, USA). Plasma F_2_-isoprostranes, as a marker of oxidative stress, were measured by gas chromatography–mass spectrometry as previously described [20].

### 2.12. Autophagy Gene Expression

Fasting whole blood was collected in Tempus™ Blood RNA Tubes (Cat#: 4342792, Applied Biosystems™, Thermo Fisher Scientific, VIC, Australia) according to the manufacturer’s instructions. Tubes were then stored at −20 °C. mRNA was later isolated from frozen samples using a Tempus™ Spin RNA Isolation Kit (Cat#: 4380204, Applied Biosystems™, Thermo Fisher Scientific Australia). cDNA reverse transcription from extracted mRNA and gene expression analysis (quantitative real-time polymerase-chain reaction) were conducted as described elsewhere [21]. Five autophagy-associated genes in white blood cells were measured, including *LC3* (microtubule associated protein 1 light chain 3 alpha, Hs01076567_g1), *BECN1* (beclin 1, Hs01007018_m1), *P62* (sequestosome 1, Hs01061917_g1), *LAMP2* (lysosomal associated membrane protein 2, Hs00174474_m1) and *TFEB* (transcription factor EB, Hs01065085_m1). As described previously [22], the relative expression of each gene was normalized for the mean of *ACTB* (actin beta, Hs01060665_g1) and *HPRT1* (hypoxanthine phosphoribosyl transferase 1, Hs02800695_m1).

### 2.13. Calculations and Statistical Analysis

HOMA-IR and HOMA-β were calculated from fasting plasma glucose and fasting plasma insulin according to the equations as described previously [23]. AUCs for glucose and insulin during the OGTT were determined using the trapezoidal rule [24]. The Matsuda index was calculated for insulin sensitivity estimation [25]. The normality of data distribution was assessed by the Shapiro-Wilk test, and data were log10-transformed if not normally distributed. Between-group differences in primary, secondary outcome variables were performed by Analysis of Covariance (ANCOVA) to test the main effects of the intervention (AvX compared with Placebo) after 12 weeks with sex and baseline values (week 0) as the covariate. Within-group effects were evaluated by paired-sample *t*-test. A post-hoc sensitivity analysis was performed on all variables additionally adjusted for percent weight change. We also performed a post-hoc sensitivity analysis selecting those participants who had insulin AUC above the median of baseline values, and removed participants > 2 SD from the cohort mean according to the Chauvinet’s Criteria [26] (n = 10 in AvX, n = 13 in placebo). All statistical analyses were performed using R programming (Version 3.6.1) and its interactive software Jamovi Version 2.3.18 [27,28], with the significance level set at 0.05.

## 3. Results

Baseline characteristics by group are shown in Table 1. The study flow diagram is shown in Figure 1, with 29 participants randomised to AvX and 31 to placebo. There were 51 participants (AvX, n = 25; Placebo, n = 26) who completed the 12-week intervention and analysed, of which 11 and 10 participants showed features of the metabolic syndrome in the AvX and placebo groups, respectively. The reasons for withdrawal are listed in the Figure 1. According to pre-study analysis of the extract, the MH concentration was 1.9%, which equated to a daily consumption of ~190 mg MH, or roughly 2.0 mg/kg, which met the target dose of 2 mg/kg.

In the completers analysis, there were no between-group effects detected for the primary outcome, glucose AUC (Table 2, Figure 2A). There were no between-group effects detected for insulin AUC (−6.38 ± 6.12 mU/L/hour vs. 0.43 ± 3.24 mU/L/hour, *p* = 0.091, Table 2, Figure 2B). However, the result was strengthened after adjusting for percent weight change (*p* = 0.056). There were no between-group differences for any other outcomes, including lipids, hsCRP, plasma F2-isoprostanes or autophagy (Appendix A) markers. However, within-group changes from baseline were observed in the AvX group only for fasting triglycerides (−0.15 ± 0.10 mmol/L, *p* = 0.035) and SBP (−4 ± 2 mmHg, *p* = 0.042), and body weight (0.6 ± 0.3 kg, *p* = 0.024) (Table 2). In the post-hoc subgroup analysis (participants above the median at baseline, n = 23), reduction in insulin AUC in the AvX group did not differ from the placebo group (−29.4 mU/L/hour vs. −5.97 mU/L/hour, *p* = 0.081), but a between-group effect was detected in the percent change in insulin AUC (16.9 ± 6.9% greater reduction in AvX vs. placebo, *p* = 0.024, Figure 3). 

There were no differences between groups for reported change in energy intake (*p* = 0.430), total protein (*p* = 0.415), carbohydrate (*p* = 0.993) or fat (*p* = 0.421) intake or activity at week 12 (Appendix A).

## 4. Discussion

This double-blinded, randomised clinical trial examined the effects of an extract of unripe avocado that was naturally enriched in MH on glucose tolerance, insulin response, lipids, hsCRP and F2-isoprostanes as markers of oxidative stress, and autophagy in nondiabetic individuals with obesity. The primary outcome measure, glucose AUC, was not different between groups. However, there was a trend towards lower insulin AUC versus placebo, after adjusting for weight change. The exploratory subgroup analysis showed reduced insulin AUC, providing preliminary evidence consistent with animal studies that the supplement reduced insulin requirements in those individuals at high risk for cardiometabolic disease.

The findings of the current study add to the limited evidence pertaining to the impact of MH on reducing the insulin response after an OGTT. The only human study in the literature was conducted by Viktora et al. (1969) [16], who reported that administration of avocado extract enriched with MH (2.2–12.8 g per day) reduced fasting insulin in five out of eight healthy men, suggesting improved insulin sensitivity. In the present study, there was a small increase (0.6 kg) in body weight in the avocado group. That was not entirely unexpected, due to a small increase in caloric intake that occurred upon consumption of the avocado supplement. The water weight of an unripe avocado is around 80%; thus, the participants consumed ~65 g of avocado daily, or an additional ~90 kcal/d. In contrast, the placebo supplement provided less than half this (~40 kcal/d). Advising the participants to replace the supplement with another item rather than adding it to their usual intake may have strengthened the findings in our study. The post-hoc subgroup analysis showed a reduction in the changes in insulin AUC for those participants with insulin AUC above the median of baseline values (participants above the median at baseline) suggesting improved insulin sensitivity.

Given a previous finding that showed a dose-dependent effect of MH on glucose metabolism in dogs [11], our study may not have provided a dose high enough with MH to impact glycaemia during an OGTT performed the following day. The reported peak in MH in dogs occurred 6–8 h after MH intake; it was reduced at 12 h and was not present the following morning at 24 h post MH dosing [11,29]. In dogs, McKnight et al. showed that the half-life of MH was 3.7 h [29]. In the only human study (n = 8) which showed reduced insulin levels in response to MH, blood and urine samples were collected immediately before and after ingestion of the MH [16]. The ingested dose of MH ranged from 2.2 to 12.8 g per person, resulting in urinary MH from 6.7% to 16%, after the ingestion [16]. In the present study, the measured concentration of MH in the extract was 1.9%, which equates to ~190 mg per day. Participants were instructed to consume MH with their breakfast meal to have peak effects in the subsequent 12 h when food intake in humans typically occurs. The majority (76%, n = 19) of participants reported consuming the supplements at breakfast, but the remainder reported dividing the supplement over 2–3 meals. The selection of timing likely meant little MH was present during the following morning OGTT, which likely lessened our ability to detect significant effects between groups.

As many CR mimetics stimulate other protective pathways invoked by CR [30], we also examined markers of oxidative stress and autophagy. There is limited research specifically examining the effects of avocado consumption on cellular autophagy. However, studies exploring the impact of other avocado compounds on autophagy provide valuable insights. For instance, Motta et al. (2021) [31] showed that avocado oil could be used as a cytoprotective food supplement by decreasing the oxidative stress and apoptotic events induced by cortisol in undifferentiated neuroblastoma cells. While avocado extract contains other bioactive compounds, we did not detect any difference between groups in genes involved in modulation of autophagy.

Consumption of fresh avocado fruit has increased greatly in the last decades due to increased availability and relatively reduced cost, combined with purported health benefits that include reduced blood lipids [32], reduced systemic inflammation [33,34] and decreased oxidative stress [35]. Avocados are also high in monounsaturated fat, oleic acid, folate, carotenoids, β-sitosterol, tocopherols and fibre [11]. In this study, we utilised pitted unripe avocados, ground into powder. Unripe avocado is naturally enriched in MH, but we recognise that effects of MH enrichment cannot be isolated from possible effects produced by other bioactive components and nutrients in avocados. Several studies have investigated consumption of 0.5–1 whole avocado, daily, on health outcomes [16,34,36,37]. Daily consumption of a ripe avocado did not alter insulin sensitivity or glycaemic outcomes in response to an OGTT after 12 weeks of intervention in adults with overweight/obesity [34,37] or after 4 weeks in patients with type 2 diabetes [38]. In contrast, a meta-analysis of 9 RCTs showed that 38–330 g/day of daily avocado consumption reduced total cholesterol and LDL-C in people with hypercholesterolemia or metabolic syndrome [32]. In the present study, a within-group reduction in triglycerides was observed in the AvX group, which suggests the study may have been underpowered to detect any possible lipid-lowering effect.

This study has several limitations. Due to the lower-than-expected MH enrichment of the extract, we determined that individuals required 10 g AvX daily (approximately 190 mg MH), which yielded an individual dose of about 2.0 mg/kg. This intake level may have introduced some additional compliance issues and the likelihood that some beneficial outcomes may have been attributed to other bioactives in the AvX. We did not assess the bioavailable levels of MH in blood in response to consumption since they were likely to be undetectable at 24 h post-consumption. Additionally, the study was performed under free-living conditions requiring reliance upon participant compliance to the program, but this was high as shown by the returned weight of the product bags. Interpretation is limited to those individuals who are overweight or obese and nondiabetic. Moreover, a high proportion of predominately Caucasian females chose to enroll, which limits the translation to males and other ethnicities.

## 5. Conclusions

In conclusion, daily consumption of an unripe avocado extract intake for 12 weeks did not alter glucose tolerance in nondiabetic individuals with obesity. The whole cohort showed a trend towards reduced insulin AUC. A sub-cohort analysis provided preliminary evidence that unripe avocado reduced insulin requirements during an OGTT, which is in alignment with previous animal research. However, larger-scale, longer-term studies involving diverse populations and higher doses of MH are needed to confirm these preliminary findings and provide conclusive evidence as to whether there is any therapeutic benefit to unripe avocado.

## Figures and Tables

**Figure 1 nutrients-15-04812-f001:**
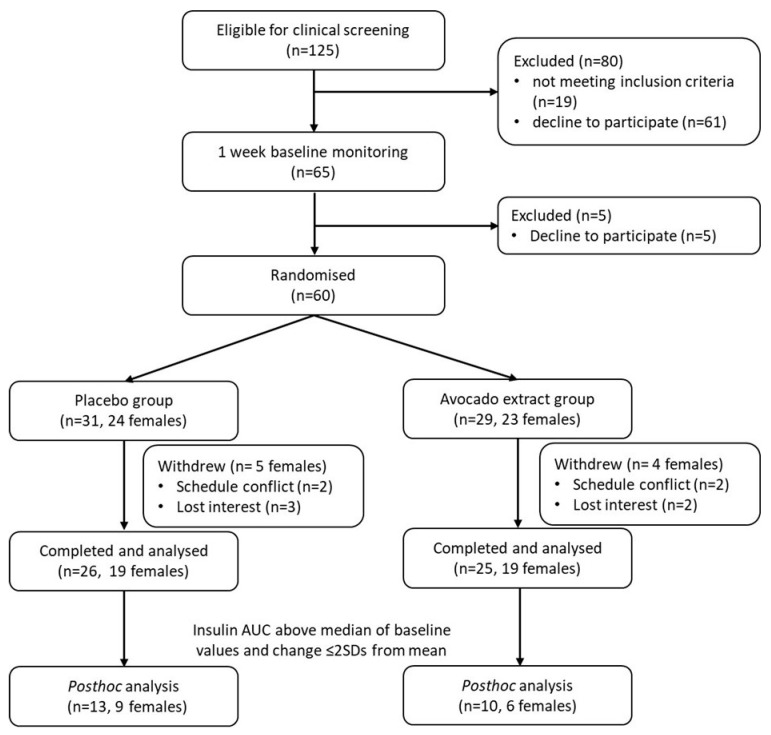
Study consort. AUC, area under the curve; SD, standard deviation.

**Figure 2 nutrients-15-04812-f002:**
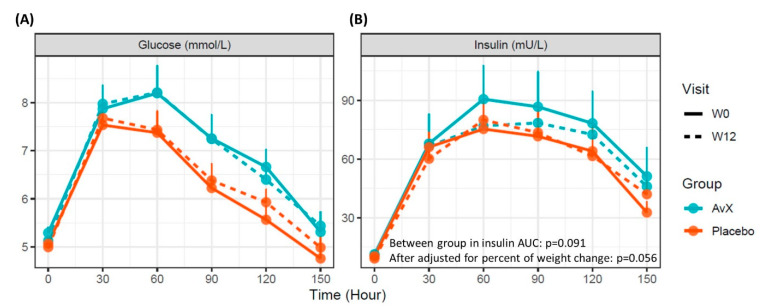
Plasma glucose (**A**) and insulin (**B**) responses over 2.5 h after the oral glucose tolerance test before and after the 12-wk avocado extract (AvX, n = 25) or placebo (n = 26) intervention in adults with obesity. Data are means ± SEMs. Between-group analysis for each variable was performed using ANCOVA, adjusted for baseline values and sex.

**Figure 3 nutrients-15-04812-f003:**
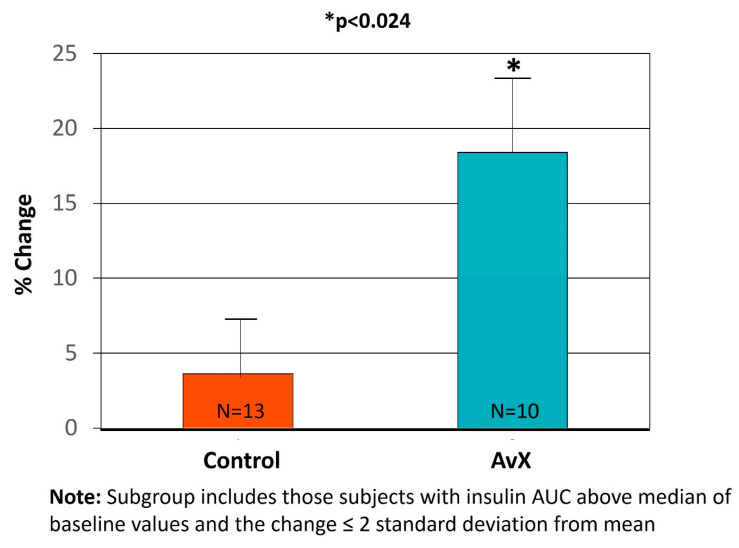
Difference in percentage of reduction in insulin AUC (cumulative insulin levels during oral glucose tolerance test) in a subgroup of participants with insulin AUC above medium at baseline.

**Table 1 nutrients-15-04812-t001:** Baseline characteristics of study participants ^1^.

	AvX (n = 29)	Placebo (n = 31)
Female/male, n	23/6	24/7
Age (Years)	51 ± 12	46 ± 13
Body weight (kg)	93.6 ± 10.0	94.5 ± 9.3
BMI (kg/m^2^)	33.8 ± 2.7	34.2 ± 2.5
Waist circumference (cm)	105 ± 10	106.0 ± 10.0
Systolic blood pressure (mmHg) ^a^	120 ± 15	114 ± 9
Diastolic blood pressure (mmHg) ^a^	77 ± 7	75 ± 6
Fasting plasma glucose (mmol/L) ^b^	5.3 ± 0.6	5.1 ± 0.6
Fasting insulin (mU/L) ^b^	10.7 ± 9.6	10.9 ± 7.5
HOMA-IR ^b^	2.6 ± 2.6	2.6 ± 2.4

^1^ Data presented as mean ± SD. ^a^, n = 30 in Placebo group; ^b^, n = 28 in AvX group and n = 29 in placebo group.

**Table 2 nutrients-15-04812-t002:** Anthropometrics, blood pressure, fasting glycaemic markers and oral glucose tolerance test indices before and after 12-wk avocado extract (AvX) or placebo intervention, including mean within-group changes and between-group differences (Δ), in adults with obesity ^1^.

	Avocado Extract Group (N = 25)			Placebo Group (N = 26)			Δ Between-Group Estimates (AvX–Placebo)	*p* Value (Between Group)
	W0	W12	Δ (W12–W0)	W0	W12	Δ (W12–W0)		
**Anthropometrics**								
Body weight (kg)	93.5 ± 2.1	94.1 ± 2.1	0.6 ± 0.3 *	95.0 ± 2.0	94.8 ± 2.1	−0.3 ±0.6	0.9 ± 0.6	0.176
BMI (kg/m^2^)	33.70 ± 0.55	33.90 ± 0.59	0.24 ± 0.09 *	34.10 ± 0.48	34.00 ± 0.55	−0.10 ± 0.21	0.36 ± 0.23	0.149
Waist circumference (cm)	105.0 ± 2.0	104.0 ± 1.9	−0.8 ± 1.2	107.0 ± 1.9	107.0 ± 2.3	0.92 ± 1.67	−2.0 ± 2.0	0.337
Body fat mass (%)	44.90 ± 1.42	44.90 ± 1.47	−0.1 ± 0.2	46.10 ± 1.33	46.30 ± 1.56	−0.1 ± 0.4	0.08 ± 0.48	0.559
SBP (mmHg)	122 ± 3	118 ± 2	−4 ± 2 *	115.00 ± 1.77	113.00 ± 2.03	−2 ± 1	0.5 ± 2.1	0.825
DBP (mmHg)	78 ± 1	77 ± 1	−1 ± 1	75.80 ± 1.19	74.70 ± 1.26	−1 ± 1	0.7 ± 1.0	0.482
**Fasting glycaemic markers**								
Fasting insulin (mU/L)	11.30 ± 1.99	10.60 ± 1.52	−0.71 ± 0.98	10.40 ± 0.86	9.17 ± 0.80	−1.2 ± 0.77	0.87 ± 1.0	0.883
Fasting glucose (mmol/L)	5.29 ± 0.12	5.13 ± 0.11	−0.16 ± 0.10	5.07 ± 0.08	5.00 ± 0.08	−0.07 ± 0.06	−0.02 ± 0.10	0.864
HOMA-beta	124.0 ± 16.4	137.0 ± 16.7	12.9 ± 11.1	141.0 ± 13.3	128.0 ± 12.1	−12.8 ± 10.7	21.4 ± 14	0.508
HOMA-IR	2.79 ± 0.54	2.49 ± 45.3	−0.30 ± 0.27	2.34 ± 0.20	17.8 ± 17.1	−0.28 ± 0.18	0.16 ± 0.24	0.506
**Oral glucose tolerance test**								
Glucose AUC (mmol/L/hour)	7.06 ± 0.36	7.02 ± 0.35	−0.04 ± 0.20	6.32 ± 0.20	6.48 ± 0.25	0.16 ± 0.14	−0.11 ± 0.25	0.678
Insulin AUC (mU/L/hour)	71.00 ± 14.30	64.60 ± 10.7	−6.38 ± 6.12	59.70 ± 7.77	60.20 ± 6.01	0.43 ± 3.24	−3.16 ± 4.82	0.091
Matsuda insulin sensitivity index	30.5 ± 74.2	24.7 ± 45.3	−5.8 ± 6.1	14.9 ±12.2	17.8 ± 17.1	2.98 ± 2.1	−2.78 ± 3.00	0.337
**Lipid profile**								
HDL-C (mmol/L)	1.43 ± 0.08	1.41 ± 0.07	−0.03 ± 0.04	1.27 ± 0.06	1.24 ± 0.06	−0.03 ± 0.03	0.03 ± 0.05	0.495
LDL-C (mmol/L)	3.38 ± 0.17	3.42 ± 0.18	0.04 ± 0.15	3.83 ± 0.19	3.70 ± 0.13	−0.13 ± 0.09	−0.02 ± 0.16	0.939
Total cholesterol (mmol/L)	5.11 ± 0.18	5.09 ± 0.19	−0.02 ± 0.16	5.41 ± 0.21	5.23 ± 0.15	−0.18 ± 0.10	−0.05 ± 0.17	0.859
Triglyceride (mmol/L)	1.48 ± 0.13	1.34 ± 0.11	−0.15 ± 0.10 *	1.55 ± 0.14	1.49 ± 0.12	−0.06 ± 0.09	−0.11 ± 0.11	0.532
**Inflammation and oxidative stress biomarker**								
hsCRP (mg/L)	2.10 ± 0.51	2.68 ± 0.65	0.58 ± 0.29	1.58 ± 0.27	1.94 ± 0.32	0.36 ± 0.20	0.18 ± 0.36	0.69
F2-Isoprostane (pmol/L)	582.0 ± 60.8	530.0 ± 53.8	−52.8 ± 63.0	624.0 ± 62.5	627.0 ± 57.8	3.2 ± 23.9	−72.6 ± 56.4	0.165

^1^ Data are presented as mean ± SEM. W, Week; BMI, body mass index; SBP, systolic blood pressure; DBP, diastolic blood pressure; HOMA-beta, homeostasis model assessment of β-cell function; HOMA-IR, homeostatic model assessment for insulin resistance; AUC, area under the curve; HDL-C, high-density lipoprotein cholesterol; LDL-C, low-density lipoprotein cholesterol; hsCRP, high-sensitive C-reactive protein. Between-group analysis for the change from Week 0 to Week 12 for each variable was performed using ANCOVA, adjusted for baseline value and sex. Within-group differences were assessed by paired-sample *t*-test (* *p* = 0.006).

## Data Availability

The data that support the findings of this study are available upon reasonable request for academic use.

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
