# Peer review of "Effects of an Unripe Avocado Extract on Glycaemic Control in Individuals with Obesity: A Double-Blinded, Parallel, Randomised Clinical Trial"

_nutrients, 2023, doi:10.3390/nu15224812_

Round 1

Reviewer 1 Report

Comments and Suggestions for Authors

The manuscript entitled „Effects of an unripe avocado extract on glycaemic control in individuals with obesity: a double blinded, parallel, randomized clinical trial” describes an RCT on the effect of the unripe avocado extract. The work sounds valid but some improvements are required.

The introduction is not exhaustive enough. Why the autophagy gene was analysed? Why the other parameters were analysed? From where was the idea of unripe avocado?

Have you checked for metabolic syndrome parameters when you recruited participants? The selected group represents obesity class II, which is a serious disease and there can be many malfunctions existing.

Why this dose of extract was selected? Maybe it was too low to show the effect?

What about other ingredients of avocado? It should be analysed. Maybe the effects observed were because of fatty acids, not the MH.

The description of the Material and Methods section is chaotic. There are few times the information about the randomization. It should not be like that.

Why the oxidative stress was analysed? It was not mentioned before.

What is a “metabolic visit”?

It should be “F2-isoprostanes”

The discussion is exaggerated. This study does not show any effect of MH, which as stated, was present in the sample in less than 2%. It has to be modified in the whole text.

The conclusion section is fine, but the conclusion in the abstract is exaggerated. Please correct.

Comments on the Quality of English Language

There are many grammatical and type errors. Sentences are not well connected. The manuscript requires editing and adding a flow.

Author Response

Dear Reviewer,

Thank you for reviewing our Manuscript ID. nutrients-2693865, entitled: " Effects of an unripe avocado extract on glycaemic control in individuals with obesity: a double blinded, parallel, randomized clinical trial". 

We thank you for your thoughtful and constructive comments. We have considered all the suggestions and incorporated them into the revised manuscript. Changes to the original manuscript are highlighted in yellow, and we believe our manuscript is stronger as a result of these modifications. An itemized point-by-point response to the reviewers’ comments is presented below. 

Yours sincerely,

Lijun Zhao, Donald K Ingram, Eric Gumpricht, Trent De Paoli, Xiao Tong Teong, Bo Liu, Trevor A Mori, Leonie K Heilbronn, and George S Roth

Reviewer 2 Report

Comments and Suggestions for Authors

The authors performed an interesting research aimed to evaluate possible glucose-lowering effects of unripe avocado-extract, known to be enriched with mannoheptulose. As a result of a double blinded, parallel, randomized clinical trial decrease of insulin production was observed, without any effects in regards of glucose concentration.

The research is well-planned and was performed at a high quality level.

Some concerns include the followings:

1.       Introduction is missing explanation why mRNA of autophagy genes were studied. Autophagy is widely explored nowadays in various aspects, but the underlying reason to concentrate at this process in this study is not justified properly. In addition, the results of mRNA analysis are very inconclusive and are presented only as supplementary material. I think that this part can be either totally omitted or further explored and presented under a different point of view. As it is obvious from the Figure S1 absence of differences may originate from the fact, that the response to AvX consumption was multidirectional: in some patients the increase of mRNA expression was observed, while other patients either demonstrated no mRNA changes or decreased autophagy mRNA expression. The factors that were associated with this heterogenic response might be further explored: which metabolic background did patients have, which basic characteristics did they have, etc. If no new results are obtained, it is better to skip this part altogether.

2.       In the methods it is indicated that patients continued intake of the prescribed medications including statins. Were groups comparable in respects of medications intake? Statins are known to induce hyperglycemia.

3.       Figure 3 – the upper text (% reduction in Insulin AUC…) should be removed from the figure itself and go to the legend, except the p-level

4.       Formatting of Table 2 is required – the first column is unreadable

Other than this, the presented research is very interesting and deserves to be published.

Author Response

(The authors gave the same response as above.)

Round 2

Reviewer 1 Report

Comments and Suggestions for Authors

All my comments have been sufficiently applied in the revised version of the manuscript.

Author Response

Dear Reviewer, 

Thank you so much for your prompt review.

Best regards,

Lijun